# Quality of sleep and associated factors among people living with HIV/AIDS attending ART clinic at Hawassa University comprehensive specialized Hospital, Hawassa, SNNPR, Ethiopia

**Asres Bedaso**[1,2]*, **Yacob Abraham**[1], **Abdi Temesgen**[3], **Nibretie Mekonnen**[4]

1 School of Nursing, College of Medicine and Health Sciences, Hawassa University, Hawassa, Ethiopia, 2 Australian Centre for Public and Population Health Research, Faculty of Health, University of Technology Sydney, Sydney, NSW, Australia, 3 Psychiatry unit, Hawassa university Comprehensive specialized Hospital, Hawassa, Ethiopia, 4 Physiology unit, Faculty of Medicine, Hawassa University College of Medicine and Health Science, Hawassa, Ethiopia

* asresbedaso@gmail.com

**Data Availability Statement:** All relevant data are within the paper and its Supporting Information files.

## Abstract

### Background

Sleep is a natural, restorative, physiological process that is characterized by perceptual disengagement from and unresponsiveness to whatever going around, which is reversible. Sleep quality refers to a sense of being rested and refreshed after waking up from sleep. People living with HIV/AIDS (PLWHA) are vulnerable to poor sleep quality as they suffer from social stigma and Anti-Retroviral drug side effects. The study aimed to examine the quality of sleep and its associated factors among people living with HIV/AIDS attending Anti-Retroviral Therapy (ART) clinic at Hawassa University comprehensive specialized hospital.

### Method

Institutional based cross-sectional study was conducted among PLWHA attending ART clinic at Hawassa University comprehensive specialized hospital from May 1–30, 2019. A systematic random sampling technique was used to select an estimated 422 study participants and data was collected using interviewer-administered technique. Sleep Quality was assessed using the Pittsburgh Sleep Quality Index (PSQI). Data were entered and analyzed using SPSS 22 software. Bivariable and multivariable logistic regression model was fitted to identify factors associated with quality of sleep. An adjusted odds ratio with a 95% confidence interval was computed to determine the level of significance with P-value less than 0.05.

### Result

Out of 422 respondents, 389 participated in the study giving a response rate of 92.1%. The prevalence of poor quality of sleep among study participants was found to be 57.6% (95%

**Funding:** The author(s) received no specific funding for this work.

**Competing interests:** The authors have declared that no competing interests exist.

**Abbreviations:** ART, Anti Retro-viral Therapy; CBE, Community Based Education; COP/ROP, Country/ Regional Operational Plan; EFV, Efavirenz; HAART, Highly Active Anti-retroviral Therapy; HADS-A, Hospital Anxiety and Depression Scale-anxiety component; HADS-D, Hospital Anxiety and Depression Scale-depression component; HIV/ AIDS, Human immunodeficiency virus or Acquired Immune deficiency syndrome; HUCSH, Hawassa University Comprehensive Specialized Hospital; NVP, Nevirapine; OSS-3, 3 Item Oslo social support scale; PLWHA, People Living With HIV/ AIDS; PSQI, Pittsburgh Sleep Quality Index; SNNPR, Southern Nations, Nationalities and Peoples' Region; SPSS, Statistical Package for Social Sciences; UNAIDS, United Nations program on HIV/AIDS; USA, United States of America; WHO, World Health Organization.

CI: 54.72, 60.48). 31.9% (124) and 30.6% (119) of study participants had anxiety and depression respectively. Being between the age of 55–64 years (AOR = 5.7, 95% CI (1.9, 17.8), Age $\geq$ 65 (AOR:6.6, 95% CI (1.2, 36.9), Monthly income <1656 Ethiopian Birr (ETB) (AOR = 2.17, 95% CI (1.06, 4.4), having anxiety (AOR = 4.4, 95% CI (2.12, 9.2), having depression (AOR = 4.97, 95% CI (2.28, 10) and poor social support (AOR = 2.9, 95% CI (1.16, 7.3) were factors associated with poor quality of sleep.

## Conclusion

The prevalence of poor quality of sleep among PLWHA was significantly high. Average monthly income, age, anxiety, depression, and social support were found to be significantly associated with poor sleep quality. Health care professionals working at the ART clinic need to assess the sleep pattern of ART clients, give psychoeducation on the prevention and management of sleep pattern problems.

## Background

Sleep is a natural, restorative, physiological process that is characterized by perceptual disengagement from and unresponsiveness to whatever going around, which must be reversible [1]. During sleep, most of the body's systems are in an anabolic state, helping to restore the immune, nervous, skeletal and muscular systems, which are vital processes that maintain mood, memory and cognitive function, and play important roles in daily functions [2, 3].

Sleep quality refers to how long an individual sleeps each night and how well he/she sleeps. Also, it includes how difficult it is for an individual to fall asleep, remain slumbering, and how many times he/she wakes up during the night. Moreover, it is a sense of being rested and refreshed after waking up from sleep [4].

Having a good sleep quality is an indicator of wellbeing whereas poor sleep quality results in increased co-morbidity, mortality, health care costs and poor quality of life of PLWHA [5]. Furthermore, poor sleep quality can cause an individual to feel tired the next day and may even be associated with long term risk of Alzheimer's disease [6].

Sleep disturbances impair the quality of life, cognitive function, and emotion of PLWHA that could lead to poor medication adherence [7]. It can induce various adverse outcomes in peoples living with HIV/AIDS (PLWHA), including diminished health-related quality of life, excessive day time sleepiness, and cognitive impairment [8]. In the HIV infected population, poor quality of sleep has been associated with, disease progression, side effects of medication, financial concerns, unemployment and inadequate knowledge about behaviors enhancing good sleep [9].

HIV/AIDS is a chronic, potentially life-threatening condition caused by the human immunodeficiency virus (HIV), which interferes with the body's ability to fight against disease causing organisms [10]. It is one of the most devastating illnesses that human beings ever faced [11]. As of 2017, approximately 36.9 million peoples worldwide are living with HIV/AIDS. Sub-Saharan Africa is the most affected region, in 2017, an estimated 66% of new HIV infections occurred in this region [12]. Ethiopia is one of Sub-Saharan counties, reported a prevalence of 1.1% HIV/AIDS infection in 2016 among individuals aged between 15–49 [13].

In the United States of America, 50–70 million adults are suffering from sleeping problems [14]. Among them, insomnia and sleep apnea commonly dominate with a prevalence of 6%-

10% and 10–25% respectively [15]. Sleep disturbances are thought to be common among HIV infected individuals in the US [16] and have a 40 to 70% prevalence of sleep disturbance and Patients with HIV were found to have a higher risk of sleep disturbances than the general population [17].

PLWHA are vulnerable to poor sleep quality as they suffer from the social stigma of the disease, unpleasant side effects of ARV medications including increased risk for metabolic syndromes [18]. Depression often considered as a logical outcome in peoples living with HIV/AIDS and consequently may lead to and exacerbates sleep disturbance [19]. Also, addictive drug use is associated with sleep disturbances in persons living with HIV/AIDS [20]. There is a controversy regarding the association of duration since HIV diagnosis with sleep quality, in which some studies state that short duration since diagnosis is negatively associated with sleep quality [17]; whereas others discussed that individuals with long durations since HIV status known are more likely to have diminished sleep quality [21, 22].

Despite all the above evidence in developed and middle-income countries, little is known in low-income countries, especially in Ethiopia. So, the current study tried to fill this gap and examined the prevalence and associated factors of poor quality of sleep among people living with HIV/AIDS attending ART clinic at Hawassa University Comprehensive Specialized Hospital.

## Methods

### Study design and setting

An institutional-based cross-sectional study was conducted to examine the prevalence and associated factors of poor quality of sleep among PLWHA attending ART clinic at Hawassa University Comprehensive Specialized Hospital (HUCSH) from May1-30, 2019. Hawassa University Comprehensive Specialized hospital is located in Hawassa city, SNNPR, which is 275 km far from Addis Ababa. Beyond other inpatient and outpatient medical services, the hospital provides ART services for PLWHA.

### Population

All PLWHA attending ART clinic at Hawassa University Comprehensive Specialized Hospital were source population. PLWHA attending ART clinic at Hawassa University Comprehensive Specialized Hospital ART clinic during the data collection period were the study population. Individual ART client attending ART clinic at Hawassa University Comprehensive Specialized Hospital was the study unit.

PLWHA attending Hawassa University Comprehensive Specialized Hospital ART clinic with age 18 years and above were included in the study but those with a severe medical condition (unconscious or critically ill) and unable to communicate due to hearing difficulty were excluded from the study.

### Sample size and sampling technique

The sample size was determined using a single population proportion formula considering assumption (Z = 1.96, d = 0.05, and P = 50%). Then, adding a 10% non-response rate, the final estimated sample size was 422. Among the total ART clients currently being enrolled in ART service, 422 study participants were selected through systematic random sampling technique using sampling fraction; K = 6. The sampling fraction (K) was obtained by dividing the total ART clients who have follow-up at Hawassa University comprehensive specialized hospital (n = 2533) by the sample size, 2533/422 which is 6. The first individual was selected using a lottery method, and the rest were selected at a regular interval (every 6th).

## Measurements and data collection technique

Data were collected by four psychiatry nurses using interviewer administered technique. During data collection daily base supervision was conducted by one MSc psychiatry professional to check for completeness of data collection tool.

Sleep quality was assessed by using the Pittsburgh Sleep Quality Index (PSQI), a 19-item self-rated scale which examined Sleep Quality and disturbances over a 1 month time interval. The tool mainly addresses seven sleep components: sleep quality, sleep latency, sleep duration, habitual sleep efficiency, sleep disturbances, use of hypnotics, and daytime dysfunction during the last month. The total PSQI score is then calculated by summing-up the seven component scores, giving an overall score ranging from 0 to 21. The score >5 points indicates poor sleep quality [23]. For the current study the internal consistency of PSQI was found Cronbach's alpha $\alpha = 0.76$.

The hospital anxiety and depression scale (HADS) was used to assess anxiety and depression. It has been validated in Ethiopia and its internal consistency was $\alpha = 0.78$ for anxiety, $\alpha = 0.76$ for depression subscales and $\alpha = 0.87$ for full scale. It has two subscales: the anxiety subscale (HADS-A) and the depression subscale (HADS-D). Each subscale contains seven items, giving a total of 14 items in the HADS. It has cutoff point $\geq 8$ for each subscale suggestive of depression and anxiety [24]. For the current study the internal consistency was Cronbach's alpha $\alpha = 0.81$ for full scale.

Social support was measured by using 3 items Oslo social support scale (OSS-3) which is classified as poor social support (3–8 OSS score), intermediate social support (9–11 OSS score), and strong social support (12–14 OSS score) [25]. For the current study the internal consistency of OSS-3 was Cronbach's alpha $\alpha = 0.73$. Current substance use: assessing the use of any substance (alcohol, tobacco, cigarette and other) in the last 3 months. Ever use of the substance was assessed if a study participant used any substance (alcohol, tobacco, cigarette and other) at least once in his lifetime.

Monthly income was categorized using the 2015 World Bank poverty line classification. World Bank re-established the international poverty line; from 1.25 US $- 1.9 US $ (1.9*29.06*30 = 1656 Ethiopian birr (ETB)) monthly income considered as the poverty line. Below poverty line: < 1656 ETB and above poverty line: $\geq$ 1656 ETB [26].

## Variables and data analysis

The dependent variable was quality of sleep (poor/good) and the independent variables were sociodemographic factors (age, sex, marital status, occupation, religion, educational status, income), clinical factors (CD4 count, WHO clinical stage of HIV/AIDS, duration since HIV/AIDS diagnosis, ART drug type, presence of co-morbid medical illness), Substance use (ever use/current use), psychosocial factors (depression, anxiety, social support) and environmental factors (noise disturbance).

Data entry and analysis was conducted using SPSS 22 software. Bivariable logistic regression analysis was conducted to identify independent factors associated with poor quality of sleep. Possible confounding (important variable which have a hidden effect on the outcome) variables were entered into a multivariable logistic regression model to identify the association of each independent variable with poor quality of sleep. In the final model, variables with a p-value of less than 0.05 declared as statistically significant, and AOR with 95% CI was calculated to determine the strength of association. Model fitness was checked using the Hosmer and Lemeshow test, and it was found to be 0.71. Multi-collinearity was checked by the variance inflation factor (VIF) and tolerance.

## Ethics approval and consent to participate

Ethical clearance was obtained from the Institutional Review Board (IRB) of Hawassa University, college of medicine and health sciences. The data collectors clearly explained the aim of the study for every study participant. Written consent was sought from individual ART client who agreed to participate. Participants who can't read and write gave a fingerprint to indicate consent. Each participant was informed that they have the right to refuse or discontinue participation at any time they want. All participants were randomly selected without any discrimination on any ground. Filled out questionnaires were carefully handled, and all access to results was kept strictly within the members of the research team.

# Results

## Socio-demographic characteristics

Out of 422 estimated study participants, 389 were included in the analysis, making a response rate of 92.2%. The mean age of respondents was 38.2 (±9.7) years. From the total study participants, 231 (59.4%) were females and 261 (67%) were married. The mean estimated monthly income of patients was 1612.7 (±1349) Ethiopian birr (ETB) (**Table 1**).

**Table 1. Socio-demographic characteristics of PLWHA attending ART clinic at Hawassa University Comprehensive Specialized Hospital, SNNPR, Ethiopia, 2019 (n = 389).**

| Variable | Category | Frequency (%) |
|---|---|---|
| Age | 18–24 | 53 (13.6) |
| | 25–54 | 233 (59.9) |
| | 55–64 | 85(21.9) |
| | >65 | 18(4.6) |
| Sex | Male | 158 (40.6) |
| | Female | 231 (59.4) |
| Religion | Orthodox | 177 (45.5) |
| | Protestant | 155 (39.8) |
| | Muslim | 51 (13.1) |
| | Others | 6 (1.5) |
| Marital status | Married | 261 (67) |
| | Single | 46 (11.8) |
| | Divorced | 38 (9.8) |
| | Widowed | 44 (11.3) |
| Educational status | Unable to read and write | 49 (12.6) |
| | Primary | 164 (42.2) |
| | Secondary | 110 (28.3) |
| | College/University | 66 (17) |
| Occupation | Civil servant | 58 (14.9) |
| | Merchant | 109 (28) |
| | Day worker | 107 (27.5) |
| | Student | 16 (4.1) |
| | House wife | 66 (17) |
| | Unemployed | 23 (5.9) |
| | Others | 10 (2.6) |
| Monthly income | <1656ETB | 245 (63) |
| | ≥1656 ETB | 144 (37) |

## Clinical characteristics

Out of 389 respondents, 267 (68.6%) were in stage I for WHO clinical staging. Regarding the time since HIV diagnosis, 314 (80.7%) have a duration above 1 year and 259 (66.6%) of respondents have CD4 count ≥500 cells/ml. 209 (53.7%) study participants used EFV based combination ART drug type, and 106(27.2%) have a co-morbid medical illness (**Table 2**).

## Substance use

Out of 389 respondents, 156 (40.1%) and 102 (26.2%) used at least one substance in their lifetime and within the last 3 months respectively (**Fig 1**).

## Psychosocial and environmental factors

Based on the Hospital Anxiety and Depression Scale, 124(31.9%) and 119(30.6%) of respondents had anxiety and depression respectively. 167(42.9%) of participants received poor social support. Out of the total respondents, 66(17%) complained about the experience of noise disturbance during their sleeping time.

## Quality of sleep

The prevalence of poor quality of sleep among PLWHA was 57.6%. The mean total PSQI score of individuals with poor quality of sleep was 8.43(±2.74) as compared to 3.21 (±1.47) of those with good quality of sleep (**S1 Table**).

**Table 2. Clinical characteristics of PLWHA attending ART clinic at Hawassa University Comprehensive Specialized Hospital, SNNPR, Ethiopia, 2019 (n = 389).**

| Clinical factors | Frequency and (%) |
|---|---|
| WHO clinical stages | |
| Stage I | 267 (68.6) |
| Stage II | 96 (24.7) |
| Stage III | 20 (5.1) |
| Stage IV | 6 (1.5) |
| Time since HIV diagnosis | |
| ≤1 year | 75 (19.3) |
| >1 year | 314 (80.7) |
| CD4 count | |
| <200 cells/ml | 11 (2.8) |
| 200–499 cells/ml | 119 (30.6) |
| ≥500 cells/ml | 259 (66.6) |
| ART drug Type | |
| EFV-based | 209 (53.7) |
| Non-EFV | 180 (46.3) |
| Presence of any comorbid chronic medical illness (DM, Ca, HTN or other) | |
| No | 283 (72.8) |
| Yes | 106 (27.2) |

**Abbreviation:** DM: Diabetes Mellitus, Ca: Cancer, HTN: Hypertension, ART: Antiretroviral Therapy, EFV: Efavirenz

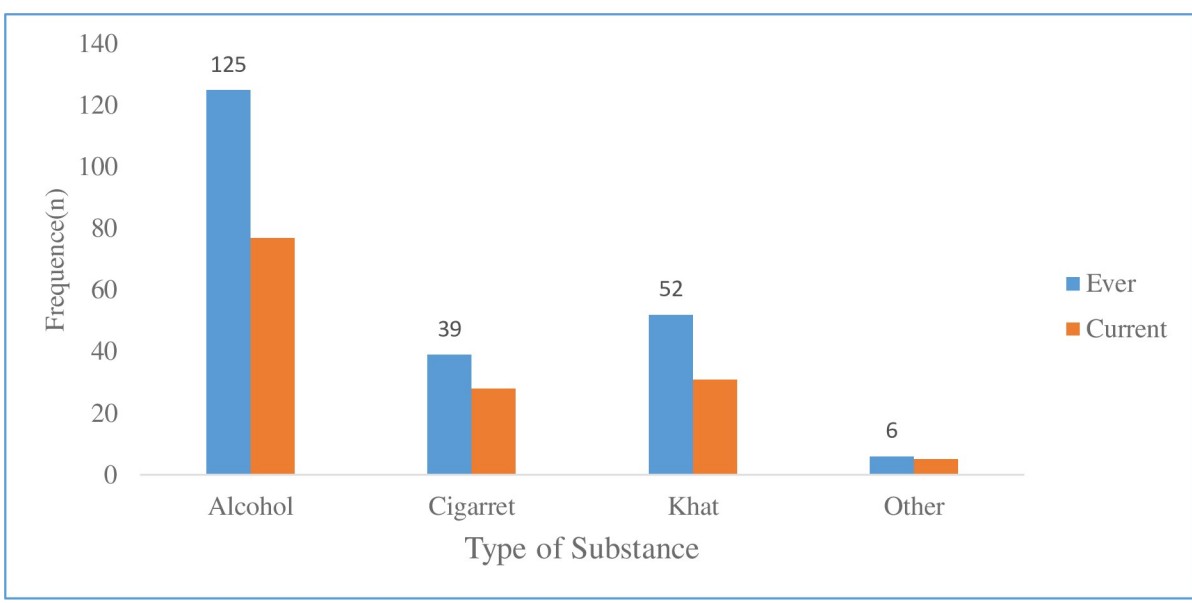

**Fig 1. Types of substances used by study participants attending ART clinic at HUCSH, SNNPR, Ethiopia, 2019 (n = 389).**

### Factors associated with poor quality of sleep

From the total variables included in the multivariable logistic regression analysis, five variables were found to be statistically significant ($P<0.05$). Accordingly, age (55–64 and >64), average monthly income (<1656 ETB), having anxiety, having depression and poor social support were found to be significantly associated with poor sleep quality (**Table 3**).

## Discussion

In the current study, the prevalence of poor sleep quality was 57.6% (95% CI: 54.72, 60.48). The result is somewhat in agreement with the result reported from Mexico which is 58.9% [22]. However, the prevalence in the current study was lower than the study conducted in Nigeria which is 59.3% [17]. The finding in the current study was higher than the result reported from South Africa which showed that the prevalence of poor sleep quality was 52% [27], 46.2% in University of Calabar teaching Hospital, South Nigeria [28], Iran 47.5% [29], China 43.1% [30], Brazil 46.7% [31], France 47% [21], Southern US 40% [32], John Hopkins Medical Institution USA 56% [33], another study in USA 46.1% [16] and Taiwan 27.4% [34]. These discrepancies might be due to various factors including differences in the geographical area, socio-cultural variation, characteristics of study participants and the inclusion and exclusion criteria used.

An individual living with HIV/AIDS age between 55–64 years was 5.7 times more likely to experience poor sleep quality compared to the younger (18–24 years) individual. Also, study participants with age ≥ 65 were 6.6 times more likely to experience poor quality of sleep compared with those with younger age groups. This implication might be related to various factors including age-related immunity deterioration, dropping of growth hormone levels, which is known to facilitate deep sleep, psychological and socio-economic factors including retirement from jobs result in a poor quality of sleep. Also, as an individual gets older, the neurological receptors that connect with sleep signaling chemicals weaken, which results in the brain face a long time figuring out when individuals are tired [35]. The current finding was consistent with the study conducted in China [36].

**Table 3. Bivariable and multivariable logistic regression of factors associated with poor quality of sleep among PLWHA attending ART clinic at Hawassa University Comprehensive Specialized Hospital, SNNPR, Ethiopia, 2019 (n = 389).**

| Variable | Category | Poor Sleep Quality | | COR (95%CI) | AOR (95%CI) |
|---|---|---|---|---|---|
| | | Yes | No | | |
| Age | 18–24 | 23 | 30 | 1 | 1 |
| | 25–54 | 114 | 119 | 1.25 (0.69,2.3) | 1.4 (0.6, 3.4) |
| | 55–64 | 72 | 13 | 7.2 (3.2, 16.1) | **5.7(1.9, 17.8)**\* |
| | ≥ 65 | 15 | 3 | 6.5(1.6, 25.2) | **6.6(1.2, 36.9)**\* |
| Marital status | Married | 142 | 121 | 1 | 1 |
| | Single | 29 | 15 | 1.6(0.86, 3.27) | 1.6 (0.5, 5.0) |
| | Divorced | 26 | 12 | 2.1(1.01, 4.46) | 1.1(0.4, 3.0) |
| | Widowed | 27 | 17 | 1.37(0.7, 2.64) | 0.9(0.34, 2.2) |
| Educational status | Unable to read & write | 36 | 13 | 2.8(1.25,6.14) | 0.5(0.13, 1.6) |
| | Primary | 101 | 63 | 1.56(0.8,2.78) | 0.6(0.26, 1.7) |
| | Secondary | 54 | 56 | 1.0(0.54,1.84) | 0.8(0.35, 2.2) |
| | College/university | 33 | 33 | 1 | 1 |
| Occupation | Civil servant | 19 | 39 | 1 | 1 |
| | Merchant | 59 | 50 | 2.4 (1.24, 4.7) | 2.7(1.01, 7.3) |
| | Day worker | 71 | 34 | 4.29(2.16,8.5) | 1.83(0.57,5.8) |
| | Student | 11 | 5 | 4.5(1.37,14.9) | 2.28(0.32,16.2) |
| | House wife | 36 | 30 | 2.46 (1.2, 5.1) | 1.27(0.37,4.3) |
| | Unemployed | 19 | 6 | 6.5 (2.2, 18.9) | 3.4 (0.9, 13.2) |
| | Others | 9 | 1 | 18.5(2.1,25.6) | 5.6 (0.3, 9.6) |
| Monthly income | <1656 ETB | 163 | 82 | 2.7(1.77,4.13) | **2.17(1.06,4.4)**\* |
| | ≥1656 ETB | 61 | 83 | 1 | 1 |
| Anxiety | No | 116 | 149 | 1 | 1 |
| | Yes | 108 | 16 | 8.6(4.86,15.4) | **4.4(2.12, 9.2)**\*\* |
| Depression | No | 119 | 151 | 1 | 1 |
| | Yes | 105 | 14 | 9.5(5.19,17.4) | **4.97 (2.28, 10)**\*\* |
| WHO stages of HIV/AIDS | Stage I | 131 | 136 | 1 | 1 |
| | Stage II | 74 | 22 | 3.49(2.05,5.9) | 0.77(0.18, 3.4) |
| | Stage III & IV | 19 | 7 | 2.82(1.14, 6.9) | 0.17(0.03, 1.2) |
| Duration since HIV diagnosis | ≤1 year | 39 | 36 | 1 | 1 |
| | >1 year | 185 | 129 | 1.41(0.85, 2.3) | 0.86(0.43, 1.7) |
| CD4 count | <200 cells/ml | 9 | 2 | 5.1(1.07,23.8) | 1.9 (0.22, 18.2) |
| | 200–499 cells /ml | 93 | 26 | 4.01(2.44, 6.6) | 3.0 (0.7, 13.2) |
| | ≥500 cells/ml | 122 | 137 | 1 | 1 |
| ART drug type | EFV-based | 136 | 73 | 1.86 (1.24,2.8) | 1.7 (0.9, 3.04) |
| | Non EFV-based | 88 | 92 | 1 | 1 |
| Chronic medical illness | No | 147 | 136 | 1 | 1 |
| | Yes | 77 | 29 | 2.45(1.5, 3.99) | 1.6 (0.8, 3.3) |
| Lifetime substance use | No | 120 | 113 | 1 | 1 |
| | Yes | 104 | 52 | 1.88(1.2, 2.87) | 1.3(0.6, 2.8) |
| Current substance use | No | 151 | 136 | 1 | 1 |
| | Yes | 73 | 29 | 2.27(1.4, 3.7) | 1.03(0.4, 2.6) |
| Social support | Poor | 127 | 40 | 5.8(3.02,11.3) | **2.9(1.16, 7.3)**\* |
| | Moderate | 77 | 90 | 1.57(0.8, 2.97) | 1.2(0.5, 2.8) |
| | Good | 19 | 35 | 1 | 1 |

\*Significant association (P-value <0.05)

\*\*Significant association (P-value <0.01), **Abbreviation:** COR: Crudes Odds Ratio, AOR: Adjusted Odds Ratio, CI: Confidence Interval, ART: Anti-Retroviral Therapy, ETB: Ethiopian Birr

Individuals below the poverty line (<1656 ETB) were 2 times more likely to experience poor quality of sleep compared with those above poverty lines. This might be related to some negative emotions posed by survival pressure over low-income earners might result in a poor quality of sleep. Also, individuals with higher socioeconomic status have been hypothesized to get positive social, psychological, and economic skills that protect against the effect of hardship [37] which would protect sleep problems. This finding is in agreement with the study reported in China [36].

Our study also revealed that the presence of both anxiety and depression increases the possibility of experiencing poor quality of sleep compared with their counterpart. This might be due to the linkages between sleep, emotional regulation and alteration in the Hypothalamic-pituitary-adrenal axis implication of psychopathology and sleep-wake cycle. Also, the presence of insomnia symptoms was higher among individuals who have anxiety and depression which results in poor sleep quality [38]. The result is in agreement with the studies conducted in the USA [16], Mexico [21] and Iran [39].

Lastly, our study result suggested, individuals who received poor social support to be 2.9 times more likely to develop poor quality of sleep. Social support is thought to promote sleep quality by providing a safe context in which close family or friends protect sleepers from enemies or other threats [40]. The current finding was in line with the study conducted in Mexico [21].

## Limitation of the study

One limitation of our study is that it's cross-sectional nature, which is weak to evaluate the cause-effect relationship. Lack of a control group limited our study's ability to characterize the sleep pattern of people with HIV/AIDS in contrast to the general population. Also, since only 1.5% of the samples in the current study were classified as stage IV, the finding can't be generalized for study participants at severe HIV stage.

## Conclusion

The prevalence of poor quality of sleep among PLWHA was high (57.6%), which indicates desperate life of individuals living with HIV/AIDS in Ethiopia. Average monthly income, age, anxiety, depression, and social support were found to be significantly associated with poor sleep quality. Health care professionals working at the ART clinic need to regularly assess the sleep pattern of ART clients, give psychoeducation on prevention and management of sleep pattern problems. Special consideration has to be given to those with age >55 years, having poor social support, having depression and anxiety, and individuals living below the poverty line.

## Supporting information

**S1 Table. Sleep quality and its components score of PLWHA attending ART clinic at Hawassa University Comprehensive specialized Hospital, SNNPR, Ethiopia, 2019 (n = 389).**
(DOCX)

## Acknowledgments

We would like to thank study participants, data collectors, and supervisors for their unreserved contribution during data collection. Also, we would like to forward our gratitude to Hawassa

University comprehensive specialized hospital ART clinic health care providers for their genuine support during data collection.

## Author Contributions

**Conceptualization:** Asres Bedaso, Abdi Temesgen.

**Data curation:** Asres Bedaso, Abdi Temesgen.

**Formal analysis:** Abdi Temesgen.

**Funding acquisition:** Asres Bedaso, Yacob Abraham, Abdi Temesgen, Nibretie Mekonnen.

**Investigation:** Asres Bedaso, Yacob Abraham, Abdi Temesgen, Nibretie Mekonnen.

**Methodology:** Asres Bedaso, Yacob Abraham, Abdi Temesgen, Nibretie Mekonnen.

**Software:** Asres Bedaso.

**Supervision:** Asres Bedaso, Yacob Abraham.

**Validation:** Asres Bedaso, Yacob Abraham, Abdi Temesgen, Nibretie Mekonnen.

**Visualization:** Asres Bedaso, Yacob Abraham, Abdi Temesgen, Nibretie Mekonnen.

**Writing – original draft:** Asres Bedaso, Yacob Abraham, Abdi Temesgen.

**Writing – review & editing:** Asres Bedaso, Abdi Temesgen, Nibretie Mekonnen.

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
