## [Decision Letter · Decision Letter 0]

1 May 2020

PONE-D-20-10185

Quality of sleep and associated factors among people living with HIV/AIDS attending ART clinic at Hawassa University comprehensive specialized Hospital, Hawassa, SNNPR, Ethiopia

PLOS ONE

Dear Mr. Bedaso,

Thank you for submitting your manuscript to PLOS ONE. After careful consideration, we feel that it has merit but does not fully meet PLOS ONE’s publication criteria as it currently stands. Therefore, we invite you to submit a revised version of the manuscript that addresses the points raised during the review process.

We would appreciate receiving your revised manuscript by Jun 15 2020 11:59PM. To enhance the reproducibility of your results, we recommend that if applicable you deposit your laboratory protocols in protocols.io, where a protocol can be assigned its own identifier (DOI) such that it can be cited independently in the future. For instructions see: http://journals.plos.org/plosone/s/submission-guidelines#loc-laboratory-protocols

We look forward to receiving your revised manuscript.

Kind regards,

Amir H. Pakpour, Ph.D.

Academic Editor

PLOS ONE

Reviewers' comments:

Reviewer's Responses to Questions

**Comments to the Author**

1. Is the manuscript technically sound, and do the data support the conclusions?

Reviewer #1: No

2. Has the statistical analysis been performed appropriately and rigorously? 

Reviewer #1: Yes

3. Have the authors made all data underlying the findings in their manuscript fully available?

Reviewer #1: Yes

4. Is the manuscript presented in an intelligible fashion and written in standard English?

Reviewer #1: No

5. Review Comments to the Author

Reviewer #1: The study entitled “Quality of sleep and associated factors among people living with HIV/AIDS attending ART clinic at Hawassa University comprehensive specialized Hospital, Hawassa, SNNPR, Ethiopia” assessed an important topic of sleep on people with HIV/AIDS attending ART clinic. I am impressed by the high response rate for the study and I do think that the study adds something to the current literature; especially the study investigated understudied population. However, grammatical errors and editing errors can be found in the manuscript. For example, in the Abstract sentence: Data was entered and analyzed using SPSS version 22, Adjusted odds ratio (AOR) with 95% CI, was estimated to assess the strength of associations and variables with p-value <0.05 were considered statistically significant. Several errors are easily observed: “Data were”; “SPSS version 22, adjusted odds ratio”. It is not the reviewer’s responsibility to point out all the errors and the authors may consider asking help from a native English speaker to improve the quality. Moreover, some uncommonly used abbreviations are directly used without explanation (e.g., HTN, ETB). The authors should provide a full spell-out for these abbreviations when they are firstly mentioned. I have additionally specific comments below.

1. The following two references are talking about the sleep among people with HIV/AIDS. The authors should incorporate them in the Introduction.

https://www.ncbi.nlm.nih.gov/pubmed/28735919

https://www.ncbi.nlm.nih.gov/pubmed/31739231

2. In the sentence “Among the total ART clients currently being enrolled in ART service, 422 study participants were selected using a systematic random sampling technique (K=6).” I cannot understand what K=6 means here.

3. Please incorporate the Variables section into the Data process and analysis section.

4. What is the meaning of “Daily base supervision was conducted by one mental health professional specialist”? Do the authors mean the psychiatry nurses were under supervision by mental health professional specialist to collect data? If so, what kind of supervision is it?

5. As the instruments are all self-report nature, I wonder what exactly the psychiatry nurses administer the survey.

6. Please provide the internal consistency information of the instruments (PSQI, HADS, and OSS-3) for the present study sample.

7. I wonder whether the PSQI and the OSS-3 have an Ethiopia version. The authors have explicitly mentioned that Ethiopia version of HADS, but not for the PSIQ and OSS-3.

8. In Data processing and analysis section, please define what a possible confounder variable is.

9. In the first paragraph of the Discussion section, I wonder whether all the compared countries used the same instrument with the same cutoff point (i.e., using PSQI with a cutoff at 5) to indicate the poor sleep quality. Also, it is unclear whether all the mentioned studies assessing people with HIV/AIDS. Please clarify.

10. Avoid using “besides” throughout the manuscript.

11. Please comment on how the study results can be generalized to other population with HIV/AIDS. Specifically, the sample was relatively healthy in the HIV stages; therefore, the results may not be applicable to people with severe HIV stage.

12. Please comment on how the prevalence rate of poor sleep quality found in the present study can be generalized to the Ethiopia. Also, please revise the sentence “Also, it is better if HUCSH ART staffs carefully assess the sleeping pattern of PLWHA and special consideration has to be given to those with age >55 years, having poor social support, having depression and anxiety and those living below the poverty line.” I think that the authors should not narrow down their implications to only one hospital. I understand that the authors may not want to overgeneralize their results; however, implication for a hospital only does not provide impact to the healthcare field.

6. PLOS authors have the option to publish the peer review history of their article (what does this mean?). If published, this will include your full peer review and any attached files.

Reviewer #1: No

---

## [Author Response · Author response to Decision Letter 0]

9 May 2020

Author’s response to reviews

Manuscript number: PONE-D-20-10185

Title: Quality of sleep and associated factors among people living with HIV/AIDS attending ART clinic at Hawassa University comprehensive specialized Hospital, Hawassa, SNNPR, Ethiopia

Date of revision: 10.05.2020 (First revision)

Authors: 

Asres Bedaso (asresbedaso@gmail.com)

Yacob Abraham: yacobabraham12@gmail.com

Abdi Temesgen: (abdipsychia@gmail.com) 

Nibretie Mekonnen: (nibretiemekonnen01@gmail.com) 

Dear Editor and Reviewers

We would like to thank the Editorial team and the reviewer for their valuable and very helpful comments. We have tried to address comments carefully and have made numerous changes to the manuscript which we hope meet an approval. We are also happy to make any further changes and additions. Please find below a copy of our response to the reviewer's comments on our manuscript reference number PONE-D-20-10185.

Kind regards,

Asres Bedaso (corresponding author) 

Reviewer #1 

1. The study entitled “Quality of sleep and associated factors among people living with HIV/AIDS attending ART clinic at Hawassa University comprehensive specialized Hospital, Hawassa, SNNPR, Ethiopia” assessed an important topic of sleep on people with HIV/AIDS attending ART clinic. I am impressed by the high response rate for the study and I do think that the study adds something to the current literature; especially the study investigated understudied population. However, grammatical errors and editing errors can be found in the manuscript. For example, in the Abstract sentence: Data was entered and analyzed using SPSS version 22, Adjusted odds ratio (AOR) with 95% CI, was estimated to assess the strength of associations and variables with p-value <0.05 were considered statistically significant. Several errors are easily observed: “Data were”; “SPSS version 22, adjusted odds ratio”. It is not the reviewer’s responsibility to point out all the errors and the authors may consider asking help from a native English speaker to improve the quality. Answer: Thanks for the comments. We have addressed the mentioned spelling and grammatical errors. Also, we have addressed other spelling and grammatical errors in the manuscript. We have used online spelling and grammar software to correct all spelling and grammar related problems. 

2. Moreover, some uncommonly used abbreviations are directly used without explanation (e.g., HTN, ETB). The authors should provide a full spell-out for these abbreviations when they are firstly mentioned. I have additionally specific comments below.

Answer: Thanks for the comments, we have addressed the problem in the revised manuscript.

3. The following two references are talking about the sleep among people with HIV/AIDS. The authors should incorporate them in the Introduction.

https://www.ncbi.nlm.nih.gov/pubmed/28735919

https://www.ncbi.nlm.nih.gov/pubmed/31739231

Answer: I would really thank the reviewer for suggesting the literatures. We have included the literatures in the revised manuscript. 

4. In the sentence “Among the total ART clients currently being enrolled in ART service, 422 study participants were selected using a systematic random sampling technique (K=6).” I cannot understand what K=6 means here.

Answer: Thanks again for the comment, we have clearly explained it in the revised manuscript. 

5. Please incorporate the Variables section into the Data process and analysis section.

Answer: Thanks, we incorporated your comment in the revised manuscript. 

6. What is the meaning of “Daily base supervision was conducted by one mental health professional specialist”? Do the authors mean the psychiatry nurses were under supervision by mental health professional specialist to collect data? If so, what kind of supervision is it?

Answer: See the revised manuscript, we have addressed your comment. 

7. As the instruments are all self-report nature, I wonder what exactly the psychiatry nurses administer the survey.

Answer: We have used interviewer administered technique as a data collection technique, not self-administered technique. This was one of the reason for the high response rate. 

8. Please provide the internal consistency information of the instruments (PSQI, HADS, and OSS-3) for the present study sample. 

Answer: Thanks for the comment, we have included the internal consistency information for PSQI, HADS and OSS-3. See the revised manuscript. 

9. I wonder whether the PSQI and the OSS-3 have an Ethiopia version. The authors have explicitly mentioned that Ethiopia version of HADS, but not for the PSIQ and OSS-Answer: PSQI and OSS-3 was not validated in Ethiopia but we have translated the English version of both tools in to Amharic by language experts. Also back translation was done by language experts who can speak both Amharic and English. Finally before the final data collection, pre-test was done to see the internal consistency of the tools and to see if there was any collinearity. 

10. In Data processing and analysis section, please define what a possible confounder variable is.

Answer: Thanks for the comment. We have defined the confounding variable in the revised manuscript. 

11. In the first paragraph of the Discussion section, I wonder whether all the compared countries used the same instrument with the same cutoff point (i.e., using PSQI with a cutoff at 5) to indicate the poor sleep quality. Also, it is unclear whether all the mentioned studies assessing people with HIV/AIDS. Please clarify.

Answer: The studies we included in discussion section used similar tool to assess Sleep quality and similar cut of point. 

12. Avoid using “besides” throughout the manuscript.

Answer: We have corrected based on your comment. See the revised manuscript. 

13. Please comment on how the study results can be generalized to other population with HIV/AIDS. Specifically, the sample was relatively healthy in the HIV stages; therefore, the results may not be applicable to people with severe HIV stage.

Answer: Thanks again for the delightful insight and suggestions. Since only 1.5% of the samples were classified as stage IV, it is difficult to generalize for those at severe HIV stage, rather we preferred to mention this as a limitation. See the revised manuscript. 

14. Please comment on how the prevalence rate of poor sleep quality found in the present study can be generalized to the Ethiopia. 

Answer: See the revised manuscript. We have made revisions in our conclusion. 

15. Also, please revise the sentence “Also, it is better if HUCSH ART staffs carefully assess the sleeping pattern of PLWHA and special consideration has to be given to those with age >55 years, having poor social support, having depression and anxiety and those living below the poverty line.” 

Answer: We have made modifications based on your comments. You can see the revised manuscript. 

16. I think that the authors should not narrow down their implications to only one hospital. I understand that the authors may not want to overgeneralize their results; however, implication for a hospital only does not provide impact to the healthcare field.

Answer: We have made some revision on the mentioned issue. See the revised manuscript. 

Cheers!!

Asres Bedaso

---

## [Decision Letter · Decision Letter 1]

14 May 2020

Quality of sleep and associated factors among people living with HIV/AIDS attending ART clinic at Hawassa University comprehensive specialized Hospital, Hawassa, SNNPR, Ethiopia

PONE-D-20-10185R1

Dear Dr. Bedaso,

We are pleased to inform you that your manuscript has been judged scientifically suitable for publication and will be formally accepted for publication once it complies with all outstanding technical requirements.

With kind regards,

Amir H. Pakpour, Ph.D.

Academic Editor

PLOS ONE

Additional Editor Comments (optional):

Reviewers' comments:

Reviewer's Responses to Questions

**Comments to the Author**

1. If the authors have adequately addressed your comments raised in a previous round of review and you feel that this manuscript is now acceptable for publication, you may indicate that here to bypass the “Comments to the Author” section, enter your conflict of interest statement in the “Confidential to Editor” section, and submit your "Accept" recommendation.

Reviewer #1: All comments have been addressed

2. Is the manuscript technically sound, and do the data support the conclusions?

Reviewer #1: Yes

3. Has the statistical analysis been performed appropriately and rigorously? 

Reviewer #1: Yes

4. Have the authors made all data underlying the findings in their manuscript fully available?

Reviewer #1: Yes

5. Is the manuscript presented in an intelligible fashion and written in standard English?

Reviewer #1: Yes

6. Review Comments to the Author

Reviewer #1: The authors have well addressed all my previous comments. I am satisfied with the revision and glad to recommend publication.

7. PLOS authors have the option to publish the peer review history of their article (what does this mean?). If published, this will include your full peer review and any attached files.

Reviewer #1: No

---

## [Editor Report · Acceptance letter]

19 May 2020

PONE-D-20-10185R1 

Quality of sleep and associated factors among people living with HIV/AIDS attending ART clinic at Hawassa University comprehensive specialized Hospital, Hawassa, SNNPR, Ethiopia 

Dear Dr. Bedaso:

I am pleased to inform you that your manuscript has been deemed suitable for publication in PLOS ONE. Congratulations! Your manuscript is now with our production department. 

With kind regards,

on behalf of

Dr. Amir H. Pakpour 

Academic Editor

PLOS ONE